

# The fecal microbiota of Thai school-aged children associated with demographic factors and diet

Lucsame Gruneck[1], Eleni Gentekaki[1,2], Kongkiat Kespechara[3], Justin Denny[4], Thomas J. Sharpton[5,6], Lisa K. Marriott[4], Jackilen Shannon[4] and Siam Popluechai[1,2]

[1] Gut Microbiome Research Group, Mae Fah Luang University, Muang, Chiang Rai, Thailand
[2] School of Science, Mae Fah Luang University, Muang, Chiang Rai, Thailand
[3] Sooksatharana (Social Enterprise) Co., Ltd., Muang, Phuket, Thailand
[4] OHSU-PSU School of Public Health, Oregon Health & Science University, Portland, OR, United States of America
[5] Department of Microbiology, Oregon State University, Corvallis, OR, United States of America
[6] Department of Statistics, Oregon State University, Corvallis, OR, United States of America

Corresponding author
Siam Popluechai, siam@mfu.ac.th

## ABSTRACT

**Background**. Birth delivery method and breastfeeding practices contribute to microbiota colonization. Other factors including diet and demographic factors structure the gut microbiome assembly and diversity through childhood development. The exploration of these factors, especially in Southeast Asian children, remains limited.

**Methods**. We investigated the fecal microbiota of 127 school-aged children in Thailand using quantitative PCR (qPCR) to assess the influence of diet and demographic factors on the gut microbiota. Multivariate analysis (multiple factor analysis (MFA) and Partial Least Squares Discriminant Analysis (PLS-DA)) were used to link particular gut microbes to diet and demographic factors.

**Results**. Diet and demographic factors were associated with variation among gut microbiota. The abundance of *Gammaproteobacteria* increased in children with infrequent intake of high fat foods. Obese children possessed a lower level of Firmicutes and *Ruminococcus*. *Bifidobacterium* was enriched in pre-teen aged children and detected at lower levels among formula-fed children. *Prevotella* was more abundant in children who were delivered vaginally. While ethnicity explained a small amount of variation in the gut microbiota, it nonetheless was found to be significantly associated with microbiome composition.

**Conclusions**. Exogenous and demographic factors associate with, and possibly drive, the assembly of the gut microbiome of an understudied population of school-aged children in Thailand.

## INTRODUCTION

Microbial colonization of the gut starts upon birth, and the composition of the microbiota community diversifies throughout childhood. Proteobacteria and Actinobacteria dominate

the gut microbiome early in life (*Zhuang et al., 2019*). As infants develop, their gut microbiota community becomes more complex and, at two to three years of age, its structure and composition begins to more closely resemble that of adults (*Rodríguez et al., 2019*). Shifts in the microbiota composition occur during this process and are influenced by mode of delivery, feeding type, and diet (*Martin et al., 2016*; *Rutayisire et al., 2016*; *Iddrisu et al., 2021*).

Early in life, the assembly of the gut microbiome highly depends on delivery mode and feeding practices (*Li et al., 2020*; *Mitchell et al., 2020*; *Coker et al., 2021*). The microbiome of infants born vaginally are more enriched in *Bifidobacterium* and/or *Bacteroides* compared with those infants delivered by cesarean section (*Yang et al., 2019*; *Reyman et al., 2019*; *Niu et al., 2020*). Over time, the association between gut microbiota and mode of delivery weakens (*Rutayisire et al., 2016*). Nonetheless, differences in the abundance of microbiota between children of different modes of delivery persists in school-aged children (*Salminen et al., 2004*). *Bifidobacterium* dominates the gut of infants receiving breast milk as compared to formula fed children (*Van den Elsen et al., 2019*; *Lawson et al., 2020*). The impact of breastfeeding duration persists later in life (*Zhong et al., 2019*). This suggests that delivery mode and feeding type could have a long-term impact on the diversity of one's gut microbiome.

Additional host-related factors such as ethnicity, age, and body mass index (BMI) contribute to alterations of gut microbiota. Geographical factors and ethnicity significantly affect gut microbiota profiles of school-aged children at the genus level (*Liu et al., 2020*). Although many studies have already monitored compositional changes in the gut microbiota depending on one's age (*e.g.*, comparing between childhood and adulthood) (*Derrien, Alvarez & De Vos, 2019*), data on microbiota profiles among school-aged children remains limited (*Odamaki et al., 2016*). Variation in the microbiome among children has also been linked to BMI (*Bervoets et al., 2013*; *Da Silva, Monteil & Davis, 2020*; *Shin & Cho, 2020*). These changes have been associated with diets which are precursors to weight gain and shape the gut microbiome (*Voreades, Kozil & Weir, 2014*; *Cho, 2021*).

Diet can shape the human gut microbiota (*Singh et al., 2017*; *Zmora, Suez & Elinav, 2019*). In children, diet explains most of the variation in gut microbiota profiles between countries or continents (*De Filippo et al., 2010*; *Nakayama et al., 2015*) as already observed in adults (*Yasir et al., 2015*; *Escobar et al., 2015*; *Ghosh et al., 2020*). Moving away from traditional diets with a high concentration of fiber, fruits and vegetables towards a Western diet rich in animal protein, fat and sugar is a cause of concern as high fat diets have been shown to disrupt the balance of gut microbiota in animal models (*Kim et al., 2012*). This effect has also been observed in humans where a decrease in the abundance of butyrate-producing bacteria has been noted in populations consuming higher-fat diets (*Wan et al., 2019*). Recently, Southeast Asian populations have begun to adopt the Western diet (*Ooraikul, Sirichote & Siripongvutikorn, 2008*). However, only a few studies have investigated the effects of this dietary pattern on the gut microbiota of young Southeast Asians (*Nakayama et al., 2017*; *Golloso-Gubat et al., 2020*).

This study is the first to examine the impact of diet and demographic factors (gender, age, BMI $z$-score, birth records, feeding type, and ethnicity) on the gut microbiota of

school-aged children in Thailand. Multivariate analyses were implemented to determine the potential contribution of multiple factors on variations of microbiota profiles as well as identifying most relevant features (microbiota taxa) for each host variable. Our results provide a preliminary overview of the associations observed between the abundance of gut microbiota and investigated factors in school-aged children from Thailand.

## MATERIALS & METHODS

### Ethics approval
All participants provided written informed consent (File S1) and the study was approved by the Ethics committee of Mae Fah Luang University (Ethics Registry: REH-61204). The study was conducted in accordance with the Declaration of Helsinki.

### Study population and group definition
We recruited 127 children from Ban Huai Rai Samakee elementary school in Chiang Rai, Thailand. The recruitment of subjects was conducted by voluntary participation through the school's administration. Parents provided informed consent prior to participation. Demographic data collection included gender, age, weight, height, ethnicity, history of birth delivery mode and feeding practice (representing the feeding mode in infancy) (File S2). The child's weight and height were measured by class instructors. Information on birth delivery method and feeding type were collected through child self-report and/or parental-report surveys. Body mass index (BMI) derived from the weight (kg) and height (m$^2$) ratio was converted into gender-specific $z$-scores for BMI-for-age according to BMI cut-offs for children (5–19 years) set by World Health Organization (De Onis et al., 2007). $Z$-scores for BMI-for-age were classified into 5 groups: severe thinness (SVThinness; $< -3$ SD; $n = 1$), thinness ($\geq -3$ SD to $< -2$ SD; $n = 5$), normal weight ($\geq -2$ SD to $+ \leq +1.0$ SD; $n = 83$), overweight (OV; $> +1$SD to $\leq +2$SD; $n = 20$), and obese (OB; $> +2$ SD; $n = 18$) (Fig. S1). Age groups were defined according to interquartile range (IQR: 25%, 50%, and 75%): age_A ($\leq 8.05$ years; $n = 32$), age_B ($8.05 <$ years $< 11.06$; $n = 61$), and age_C ($\geq 11.06$ years; $n = 34$). Five ethnic groups were recorded in this study: Akha ($n = 39$), Chinese ($n = 34$), Lahu ($n = 5$), Thai ($n = 19$), and Thai Yai ($n = 30$). Birth delivery mode comprised vaginal delivery ($n = 85$) and cesarean section ($n = 42$). Feeding types were categorized into three groups: breastfeeding ($n = 98$), formula feeding ($n = 20$), and mixed feeding ($n = 9$).

### Dietary information
Dietary habits of children were surveyed using a Thai short dietary behaviors screener developed by *Let's Get Healthy!* for use in Thai ("LGH20 Food Behaviors Screener, Thai"; OHSU Institutional Review Board protocol #3694). The screener included 20 questions that grouped participants across five dietary behavior categories: Healthy eating behavior (HEB), fruits and vegetables (FV), high sugar foods and beverages (HSFB), high salt foods (HSF), and high fat foods (HFF) (File S3A). Answer options measuring frequency of consumption were divided into four levels: Frequently (daily), sometimes (weekly), infrequently (monthly), and never. The scores for HEB and FV were assigned as 3 (daily),

2 (weekly), 1 (monthly), or 0 (never). The responses for HSFB, HSF, and HFF were reverse scored. Total component scores (*i.e.,* a sum score for each category) were divided into quartiles to assign levels of risk (low, low to moderate, moderate to high, and high) (Files S3B and S3C). Highest frequencies of HEB and FV consumption would be associated with low risk, while high risk would characterize children eating mostly HSFB, HSF, and HFF. The instrument screens general dietary behaviors, but does not provide a quantitative assessment of portion size and frequency to permit quantification of a specific food or nutrient intake. Instead, intake rankings permit categorization of individuals according to overall dietary behaviors, such as healthy eating or high consumption of fatty foods.

## Sample collection, DNA extraction, and quantitative PCR

Fecal samples were collected from all children in sterilized containers and immediately frozen at $-80$ °C. Microbiota DNA was extracted from fecal samples using the innuPREP Stool DNA Kit (Analytik Jena Biometra, Jena, Germany) according to the manufacturer's instructions. DNA yield and purity were determined using the Take 3 Micro-Volume Plate (Biotek, Winooski, VT, USA). Absolute quantification of bacteria was then conducted by qPCR using Real-Time Thermal Cyclers CFX96 Touch™ (Bio-Rad, Singapore). Primers targeting microbiota 16s rRNA genes used in this study are summarized in Table S1. Reactions consisted of template DNA, forward and reverse primers, 1X SYBR green (2X SensiFAST™ SYBR No-ROX mix, BIOLINE, UK), and nuclease-free water. The assay conditions and calculations of microbiota copy numbers were performed according to previously described protocol (*Chumponsuk et al., 2021*). The average estimates of microbiota abundance by converting CT values were expressed as logarithmic copy number per gram of wet weight feces.

## Statistical analysis

A sum score for dietary behaviors of children was visualized as a bar plot with ggplot2 (*Wickham, 2009*). Association between dietary behavior was assessed using Spearman's rank correlation and visualized with corrplot version 0.84 (*Wei & Simko, 2017*). Normality and homogeneity of variance were tested by Shapiro–Wilk test and Levene's test (stats package version 4.0.3) (*R Core Team, 2020*). Differences in the abundance of gut microbiota (File S4) between groups (dietary behaviors and demographic factors) were determined by one-way ANOVA, Welch's $t$-test, and Kruskal-Wallis rank sum test ($p < 0.05$) followed by multiple comparisons using Tukey's HSD test, pairwise t-tests, and Dunn's test with Benjamini–Hochberg (BH) $p$-value correction (hereafter referred to as $q$-value) (stats package version 4.0.3) (*R Core Team, 2020*) and FSA package version 0.8.31 (*Ogle, Wheeler & Dinno, 2020*). Association between birth delivery mode and the abundance of gut microbiota was determined by permutational multivariate analysis of variance (PERMANOVA) with adjustment for covariates (age and feeding type). Group dispersions based on a maximum distance were measured by betadisper with 999 permutations in the R package vegan (version 2.5-6) (*Oksanen et al., 2016*). Multiple factor analysis (MFA) was performed to evaluate the influence of host variables (dietary behaviors and demographic factors) on variations of gut microbiota using FactorMine R version 2.3 (*Lê, Josse & Husson, 2008*).

Contribution of variables to the data set was visualized with Factoextra version 1.0.7 (*Kassambara & Mundt, 2020*). To investigate the most relevant features (microbiota taxa) in characterizing each host factor, Partial Least Squares-Discriminant Analysis (PLS-DA) was carried out by the mixOmics package version 6.12.2 (*Rohart et al., 2017*). Canonical mode with 100 iterations was used as a parameter for classifying classes (groups of samples). Receiver operating characteristic curve (ROC curve) and area under the curve (AUC) were also calculated to examine the validity of supervised classification results. Predicted scores of categorical outcomes were compared between one class *versus* the others by Wilcoxon test (*Rohart et al., 2017*). The classification accuracy of PLS-DA models is interpreted as follows: no discrimination (AUC 0.5), low discrimination (AUC 0.6 to 0.7), acceptable (AUC 0.7 to 0.8), excellent (AUC 0.8 to 0.9), and outstanding (AUC > 0.9) (*Lobo, Jiménez-valverde & Real, 2008*; *Mandrekar, 2010*). All analyses were performed in R software version 4.0.3 (*R Core Team, 2020*). A more detailed explanation of multivariate analyses is described in File S5.

## RESULTS

### Dietary behaviors

The frequencies of dietary behaviors of children varied greatly in their score value (Fig. S2). To determine their relationship between diet behaviors, we performed a correlation analysis based on Spearman's rank correlation coefficient. After multiple testing corrections using the Benjamini–Hochberg method, we found that high sugar foods and beverages consumption were significantly correlated with high salty foods consumption ($rho = 0.39$, $q < 0.0001$) and high fat foods ($rho = 0.25$, $q = 0.01$, Fig. S3). A positive association between high salt and high fat behaviors was also detected ($rho = 0.27$, $q = 0.01$). Moreover, the fruits and vegetables consumption were negatively correlated with every dietary behavior except for those associated with healthy eating behaviors ($rho = 0.2$, $q = 0.04$). This healthy eating behavior was negatively correlated with consumption of fatty foods ($rho = -0.23$, $q = 0.02$). Despite the strength of association being considerably weak, the results identified a trend in children reporting high unhealthy foods consumption (*e.g.*, HSFB, HSF, HFF) also reporting low healthy foods behaviors (HEB and FV).

### Gut microbiota associated with dietary behaviors

MFA constructed by integration of dietary behaviors and abundance of gut microbiota revealed variation in gut microbiota profiles of children (File S6A). *Bacteroides* was highly correlated with dimension 1 (Dim 1; $r = 0.91$, $p < 0.0001$), followed by *Gammaproteobacteria* ($r = 0.90$, $p < 0.0001$) and total bacteria ($r = 0.89$, $p < 0.0001$) (Fig. 1A). Variation in the abundances of these taxa was best explained by HFF behaviors, with an increasing trend in microbial abundances indicated in HFF-low risk (coordinate = 1.43, $p = 0.02$; Fig. 1B). In Dim 2, the clusters were separated according to the number of individuals distributed in each diet category. *Ruminococcus* ($r = -0.21$, $p = 0.02$) and *Akkermansia* ($r = -0.26$, $p < 0.01$) described the distribution of HFF-low risk in Dim 3 (coordinate = 1.83, $p < 0.0001$) and Dim 4 (coordinate = 1.46, $p < 0.001$), respectively (Fig. 1C). Both genera were decreased in individuals with low HFF behaviors (Fig. 1D).

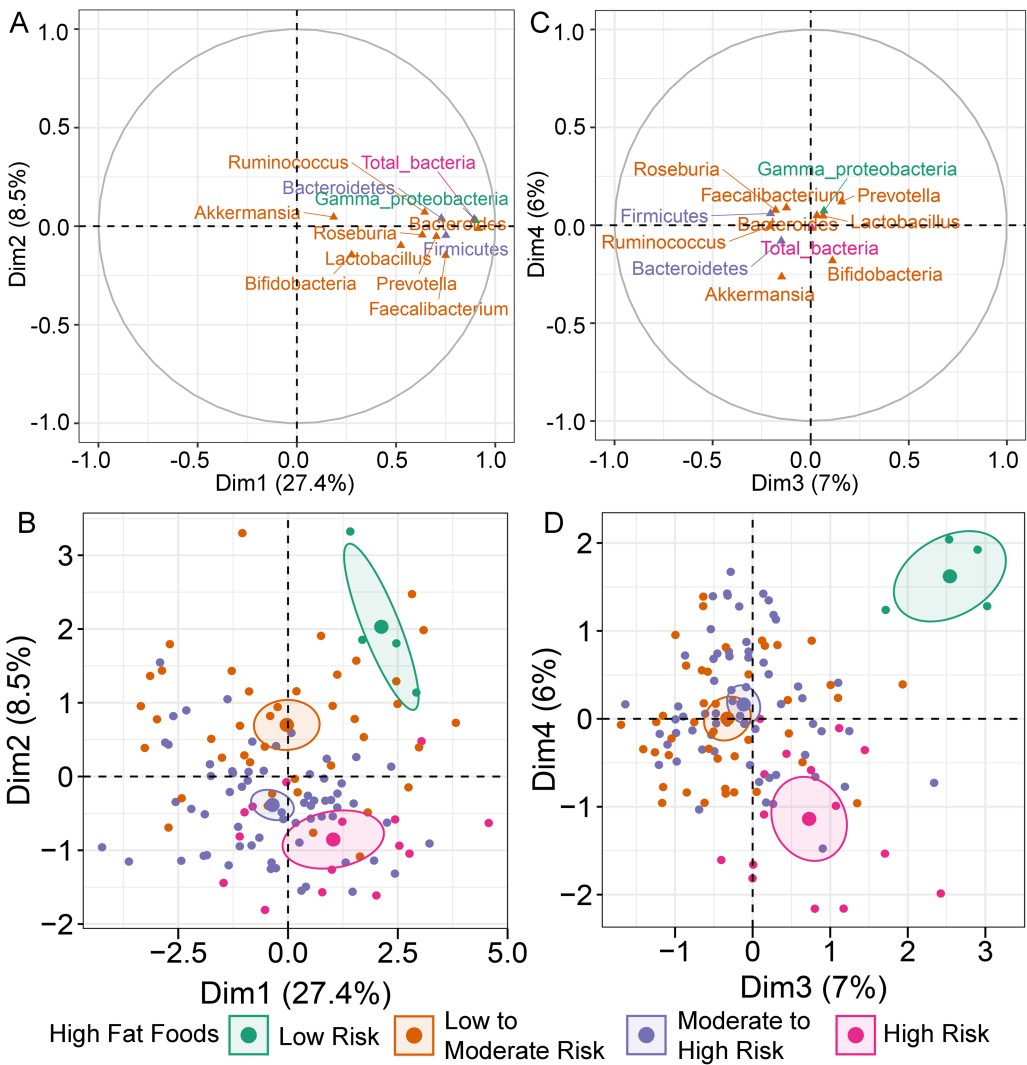

**Figure 1** **Multiple factor analysis (MFA) of the integration of gut microbiota and dietary behaviors of school-aged children.** The correlation circle plot showing the correlation between quantitative variables (microbiota taxa at the phylum, class, and genus levels) and dimensions: (A) Dim 1 and 2, (C) Dim 3 and 4. A variable that is close to the circle is highly correlated to the dimension. (B) The factor map of individual profiles grouped by high fat foods (HFF) consumption in Dim 1 and 2. (D) The factor map of individual profiles grouped by high fat foods (HFF) consumption in Dim 3 and 4. The categorical variables were specified by the 95% confidence ellipses.

Other diet behaviors (HEB, FV, HSF, and HSFB), however, had a lower coordinate on the first, third and fourth axes of the MFA factor map than HFF suggesting less contribution of these dietary behaviors to the variation in gut microbiota profiles of children in this study.

Further analysis of the association between gut microbiota and dietary behaviors using PLS-DA also identified relevant features (*i.e.,* microbiota taxa) in classifying dietary behaviors based on the level of consumption. Total bacteria and *Gammaproteobacteria*, highly contributed to discrimination of samples along component 1 (Dim 1), and

strongly characterized HFF-low risk (AUC = 0.81, $p$ = 0.04, Fig. 2A, Fig. S4A). The abundances of total bacteria ($p$ = 0.02, Fig. 2C), *Gammaproteobacteria* ($p$ < 0.0001, Fig. 2E), and *Lactobacillus* ($p$ = 0.01, Fig. 2D) were significantly different among HFF categories. After adjustment by multiple comparisons using the Benjamini–Hochberg method, *Gammaproteobacteria* significantly increased in children with low HFF compared to those with high HFF ($q$ < 0.001), moderate to high risk HFF ($q$ < 0.001), and the highest HFF consumption ($q$ = 0.03). In component 2 of PLS-DA for HFF, *Lactobacillus* and *Ruminococcus* were the most discriminative bacteria in children reporting low HFF consumption (AUC = 0.82, $p$ = 0.03, Fig. 2B, Fig. S4B). However, a significant difference in the abundance of *Lactobacillus* was detected between low HFF to moderate and high HFF consumption after adjustment ($q$ = 0.05, Fig. 2D). Moreover, PLS-DA for fruits and vegetables (FV) consumption showed that total bacteria, *Prevotella*, *Bacteroides*, and *Faecalibacterium* were the top three bacteria that separated children with high FV (FV-low risk) from those with lower FV consumption (low to moderate risk and moderate to high risk FV consumption) (Fig. S5A; AUC = 0.66, $p$ = 0.01). The abundance of total bacteria was also significantly higher in those reporting high FV as compared to those reporting lower FV consumption ($q$ = 0.04, Fig. S5C). Nevertheless, the classification was better in the second component where *Roseburia* and *Ruminococcus* contributed to high FV consumption (Figs. S5B and S5D; AUC = 0.70, $p$ < 0.001). For high salty foods (HSF), *Faecalibacterium* characterized moderate to high HSF consumption followed by *Bifidobacterium* and *Roseburia* on component 2, whereas *Lactobacillus* was associated with low HSF consumption (Fig. S6; AUC = 0.70, $p$ < 0.001). When considering healthy eating behavior (HEB) and consumption of high sugar foods and beverages (HSFB), the supervised analysis yielded no discrimination between classes (AUC < 0.6, $p$ > 0.05). Regarding the observed variability of individuals with different levels of dietary consumption, both MFA and PLS-DA analyses suggested that the consumption of high fat foods had the highest influence on the gut microbiota abundances in children.

## Associations between demographic factors and gut microbiota in children

Analysis of gut microbiota with integration of six demographic factors (gender, age, BMI $Z$-score, ethnicity, birth delivery records, and feeding type) illustrated differences of association patterns with the gut microbiota among the demographic categories (Fig. 3 and File S7). The MFA explained 18.6% and 8.3% of the variance in Dim 1 and Dim 2, respectively (Fig. S7A). *Bacteroides*, *Gammaproteobacteria*, and total bacteria were the top three variables that described individual variation in Dim 1 ($p$ < 0.0001, Fig. S7B). Their abundances decreased in underweight (Thinness) and Thai ethnicity children, while an increasing trend contributed to normal weight (Table 1, Figs. 3A and 3B). In Dim 2, *Lactobacillus* mainly described the variation of individual profiles grouped by delivery mode ($R^2$ = 0.37, $p$ < 0.0001), BMI z-score ($R^2$ = 0.34, $p$ < 0.0001), and age tertile ($R^2$ = 0.31, $p$ < 0.0001) (Figs. 3C and 3D). Abundance of *Lactobacillus* decreased in children delivered vaginally, and in those of normal weight, and oldest age (age_C) but increased in those delivered by cesarean section, OB (obese), and youngest age (age_A).

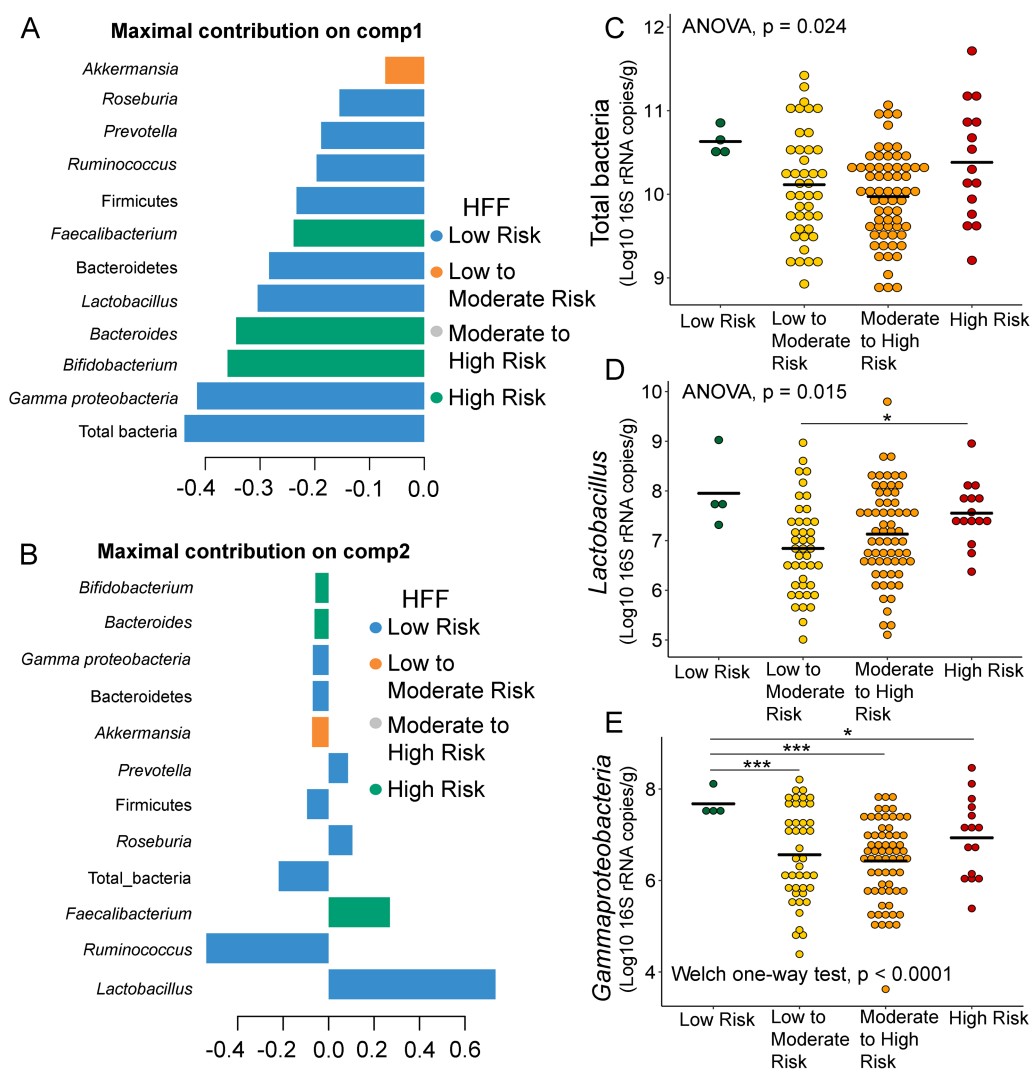

**Figure 2 Partial least squares discriminant analysis (PLS-DA) of gut microbiota in school-aged children with high fat foods (HFF) consumption.** Discriminant analysis demonstrating variable selection (microbiota taxa) for which the median (method = 'median') is maximum in component 1 (A) and component 2 (B). Horizontal bars indicate each bacterial taxon assigned to HFF consumption and their length corresponds to the loading weight. The importance of the bacteria contributing to the dimension runs from bottom to top. (C–E) Boxplots showing normalized bacterial abundances based on log10 qPCR 16S rRNA copy number per gram of feces. Asterisks indicates a significant difference in bacterial abundance among HFF consumption (***$q < 0.001$, *$q < 0.05$, Tukey's HSD test and pairwise $t$-tests with Benjamini-Hochberg $p$-value correction method).

Increased *Gammaproteobacteria* in middle age students (age_B), underweight (Thinness), and Thai ethnicity characterized Dim 3 (respectively, Figs. S8A–S8C), while this bacteria was decreased in Lahu ethnicity and oldest age (age_C). Variation of individuals in Dim 4 was mainly described by Firmicutes and ethnicity ($R^2 = 0.45$, $p < 0.0001$): the abundance of these bacteria was increased in children of Lahu and Thai ethnicity, but decreased in those of Chinese and Akha ethnicity. In Dim 5, OV (increased) had a contrasting profile of

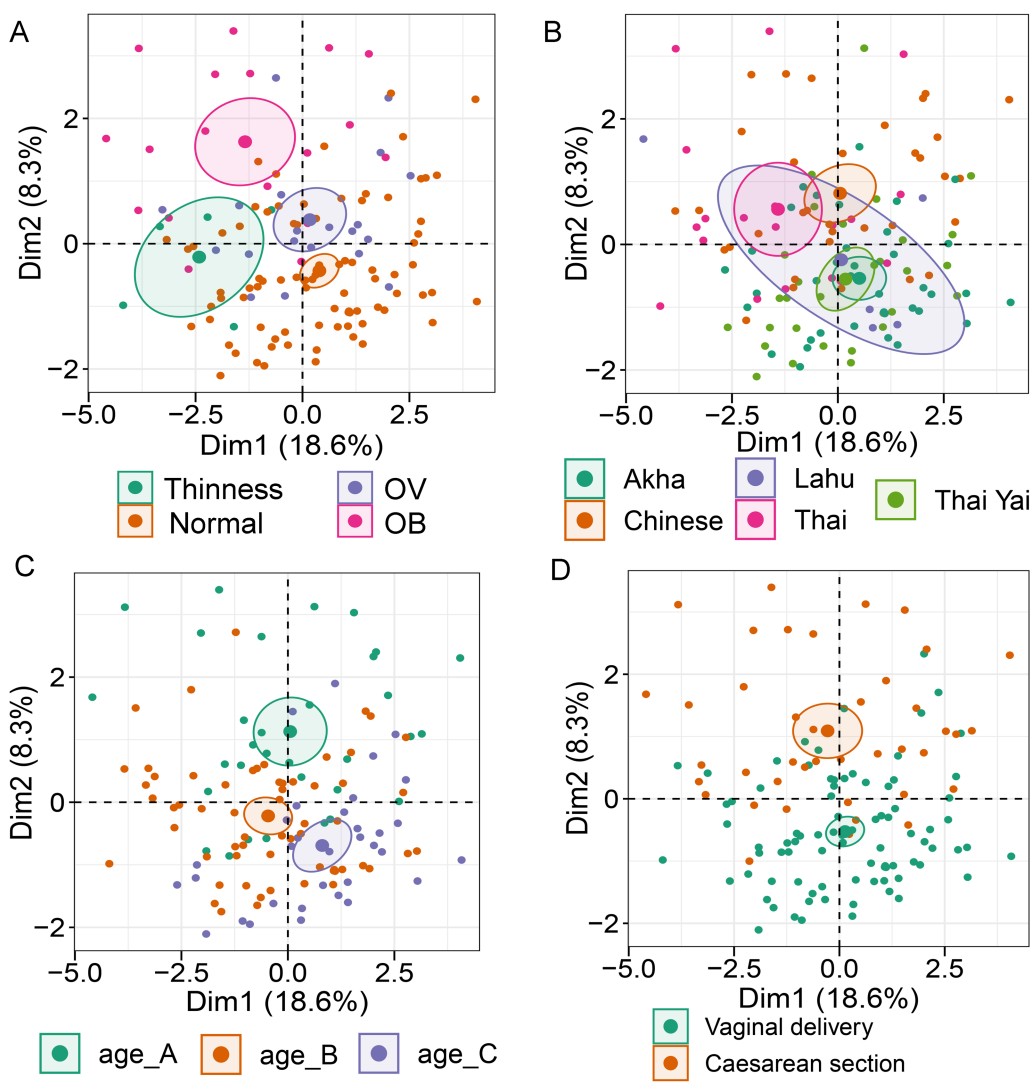

**Figure 3** **Multiple factor analysis (MFA) of the integration of gut microbiota in school-aged children and demographic factors.** The factor map of individual profiles grouped by BMI $z$-score (A), ethnicity (B), age tertile (corresponding to 25%, 50%, and 75%); age_A $\leq$ 8.05 years, age_B 8.05 < age < 11.06 years, age_C $\geq$ 11.06 years (C), and birth delivery mode (D). Individual variables were specified by the 95% confidence ellipses.

*Ruminococcus* to OB (decreased). A similar pattern of this bacterial genus was also described for mixed feeding (increased) and formula feeding (decreased) (Fig. S8D). Considering all demographic variables included in the MFA, gender had the least contribution to the variation in microbial abundances, while other factors were associated with subtle differences, which may be of relevance to profiling the gut microbiota in children.

## Correlation between gut microbiota and BMI *z*-score

Comparisons of gut microbiota across BMI *z*-score groups showed a significant difference in the abundances of Firmicutes ($p < 0.01$) and *Ruminococcus* ($p = 0.01$) (Figs. 4A–4B). After

**Table 1  Gut microbiota and categorical variables (demographic factors) that most described variation of individuals in each dimension obtained by the MFA.**

| Dimension | Dimension described by quantitative variable (bacterial taxon) | Correlation with the dimension ($r$); $p$-value[a] | Dimension described by categorical variable | [b]Coordinate; $p$-value[c] |
|---|---|---|---|---|
| 1 | *Bacteroides* | 0.90; <0.0001 | Thinness (BMI $z$-score) | −1.62; <0.01 |
| | *Gammaproteobacteria* | 0.88; <0.0001 | Thai ethnicity | −1.29; <0.001 |
| | total bacteria | 0.85; <0.0001 | Normal weight (BMI $z$-score) | 1.19; $p$<0.001 |
| | | | Vaginal delivery | −0.81; $p < 0.0001$ |
| 2 | *Lactobacillus* | 0.26; <0.01 | Normal weight (BMI $z$-score) | −0.77; $p < 0.0001$ |
| | | | [d]Age_C | −0.76, $p < 0.001$ |
| | | | Cesarean section | 0.81; $p < 0.001$ |
| | | | OB (BMI $z$-score) | 1.29; $p < 0.0001$ |
| | | | [d]Age_A | 1.06; $p < 0.0001$ |
| | | | [d]Age_B | 1.00; $p < 0.0001$ |
| 3 | *Gammaproteobacteria* | 0.21; 0.02 | Thinness (BMI $z$-score) | 2.03; $p < 0.0001$ |
| | | | Thai ethnicity | 1.58; $p < 0.0001$ |
| | | | Lahu ethnicity | −2.45; $p < 0.0001$ |
| | | | [d]Age_C | −0.92; $p < 0.0001$ |
| 4 | Firmicutes | 0.29; 0.01 | Lahu ethnicity | 1.59; $p < 0.0001$ |
| | | | Thai ethnicity | 0.85; $p < 0.0001$ |
| | | | Chinese ethnicity | −1.08; $p < 0.001$ |
| | | | Akha ethnicity | −0.97; $p < 0.01$ |
| | | | OV (BMI $z$-score) | 1.16; $p < 0.0001$ |
| 5 | *Ruminococcus* | 0.29; <0.0001 | OB (BMI $z$-score) | −1.07; $p < 0.01$ |
| | | | Mixed feeding | 1.02, $p < 0.01$ |
| | | | Formula feeding | −1.21; $p < 0.0001$ |

**Notes.**
[a]An *F*-test was used to assess whether the variable had a significant influence on the dimension.
[b]A positive value indicates an increasing trend, while a negative value represents a decreasing trend.
[c]A *t*-test was done to see whether the coordinates of the individuals in one category are significantly different from others.
[d]Age tertile (corresponding to 25%, 50%, and 75%); age_A ≤ 8.05 years, age_B 8.05 < age < 11.06 years, age_C ≥ 11.06 years.

adjustment by multiple comparisons, the abundance of Firmicutes and *Ruminococcus* were significantly higher in students of normal weight ($q < 0.01$) and OV ($q < 0.05$) compared to obese. The supervised analysis also indicated discriminations of these microbiota taxa between BMI $z$-score groups (Fig. 4C). Normal BMI was highly associated with increased abundance of *Ruminococcus* (component 1: AUC = 0.63, $p = 0.02$, Fig. 4D, Fig. S9), while low abundance of Firmicutes and *Ruminococcus* in OB discriminated them from those in other groups (component 1: AUC = 0.68, $p = 0.02$, Fig. 4D, Fig. S9). A decreasing trend in the abundance of *Gammaproteobacteria* and *Bacteroides* contributed to thinness (AUC = 0.76, $p = 0.04$, Fig. 4E), however, their association was less important.

### Relation between gut microbiota abundance with age group

Differences in the abundance of Firmicutes ($p = 0.05$) and *Bifidobacterium* ($p = 0.02$) were detected at different age tertiles of school-aged children (Fig. S10). Significant increase in Firmicutes ($q = 0.04$) was found in oldest children over 11 years of age (age_C) compared to

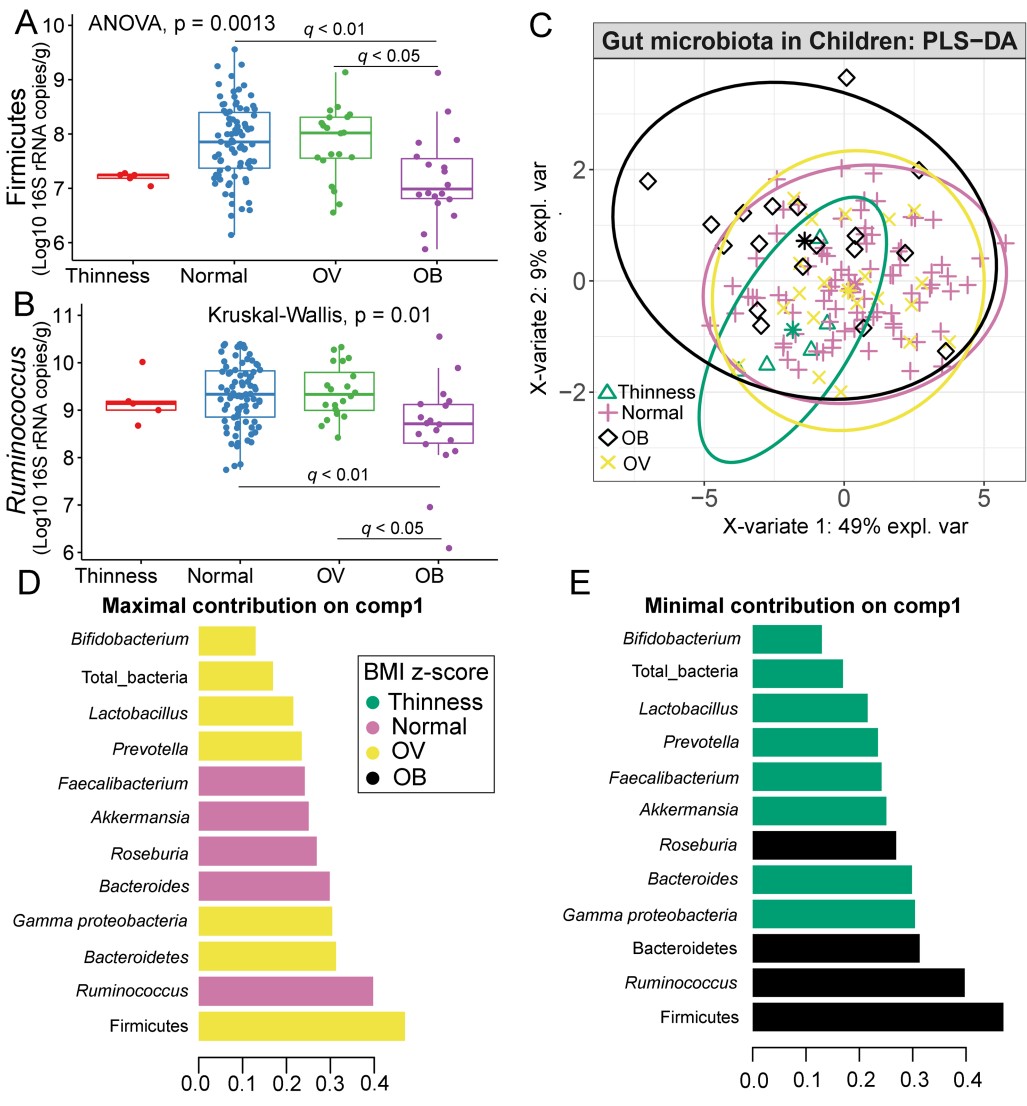

**Figure 4** **Partial least squares discriminant analysis (PLS-DA) of gut microbiota in school-aged children with different BMI z-score groups.** (A–B) Boxplots showing normalized bacterial abundances based on log10 qPCR 16S rRNA copy number per gram of feces. Significant differences in the abundance of Firmicutes and *Ruminococcus* were found between BMI $z$-score groups after adjusting $p$-values for multiple comparisons ($q < 0.05$, Tukey's HSD test and Dunn's test with Benjamini-Hochberg $p$-value correction method). (C) The sample plot represents variations in gut bacterial profiles of school-aged children with different BMI $z$-score groups (95% confidence ellipses). An explained variance was based on X-variate (normalized bacterial abundances). (D–E) Discriminant analysis demonstrating variable selection (microbiota taxa) for which the median (method = 'median') is maximum in component 1 and 2 of the sample plot. Horizontal bars indicate each bacterial taxon assigned to BMI $z$-score levels and their length corresponds to the loading weight. The importance of the bacteria contributing to the dimension runs from the bottom to the top of the figure. OV = overweight, OB = obese. BMI $z$-score cut-off points were based on WHO Multicentre Growth Reference Study Group (2006); SVThinness (severe thinness) $< -3$SD, Thinness $\geq -3$SD to $< -2$SD, Normal $\geq -2$SD to $\leq +1$SD, OV (overweight) $> +1$SD to $\leq +2$SD, OB (obese) $> +2$SD.

those in age_B ($8.05 <$ age $< 11.06$ years) (Fig. 5A). Age_C also showed greater abundance of *Bifidobacterium* than age_A ($q = 0.02$) and age_B ($q = 0.04$) groups (Fig. 5B). Further evaluation of age-associated differences in the gut microbiota of children by PLS-DA revealed certain microbiota taxa contributing to the discrimination. The PLS-DA plot displayed variations in microbiota profiles according to age tertile (Fig. 5C). Feature classification indicated Firmicutes, *Bacteroides*, *Roseburia*, *Prevotella*, and *Ruminococcus* as the top five more abundant microbiota taxa in the oldest school children (age_C) (Fig. 5D). Of these, Firmicutes had the highest contribution to age_C in component 1 (AUC $= 0.62$, $p = 0.03$, Fig. S11). The model supports that children over 11 years of age have a higher abundance of this microbiota phylum.

**Comparison of microbiota abundance in different delivery mode**

In this study, we included a record of childbirth to determine its association with the gut microbiota. A comparison of means between the two birth delivery modes showed no significant difference in their abundance of microbiota (Fig. S12). When we performed PERMANOVA with adjustment for covariates (age and feeding type; File S8), the test indicated that birth delivery mode was significantly associated with the abundance of *Prevotella* ($p = 0.03$, Fig. S13A), while no influence of sample dispersions was detected ($p = 0.08$, and Fig. S13B). Further analyses using PLS-DA also revealed variations of gut microbiota abundance based on birth delivery mode (Fig. S13C). The enrichment of *Prevotella* in vaginal delivery was clearly distinguished from that observed in those delivered by cesarean section (component 1: AUC $= 0.69$, $p < 0.001$, Figs. S13D and S13E).

**Differences in the abundance of gut microbiota of children associated with feeding type**

The gut microbiota profile of children varied across feeding types (Fig. S14). A comparison of microbiota abundances among the three feeding types (breastfeeding, formula feeding, and mixed feeding) showed significant differences in the abundance of Firmicutes and *Bifidobacterium* ($p < 0.05$). Both bacterial taxa were significantly higher in mixed feeding children than in those receiving formula feeding ($q < 0.05$, Figs. 6C, 6D). Abundance of *Bifidobacterium* was significantly increased in children breastfed as infants compared to those formula fed as infants ($q = 0.01$, Fig. 6D). We then analyzed the association between gut microbiota and feeding type using PLS-DA to identify key-discriminatory microbiota taxa. Although the PLS-DA components displayed overlapping clusters (Fig. 6A), several differentially abundant bacteria that contributed to the variation in feeding type were indicated (Fig. 6B). The classification model suggested that *Faecalibacterium* (Fig. 6E), Firmicutes, *Roseburia* and *Bifidobacterium* increased following mixed feeding in component 1 (AUC $= 0.60$, $p = 0.31$, Fig. S15A). In component 2, a similar pattern was observed for Firmicutes and *Ruminococcus* (AUC $= 0.71$, $p = 0.03$), whereas *Gammaproteobacteria* increased in formula fed children (AUC $= 0.79$, $p < 0.0001$) (Figs. S15B and S16A).

**The influence of gender towards gut microbiota profile in children**

Comparisons of the abundances of gut microbiota found no significant difference between gender (Fig. S17). This factor, however, accounted for 47% of the variation in microbial

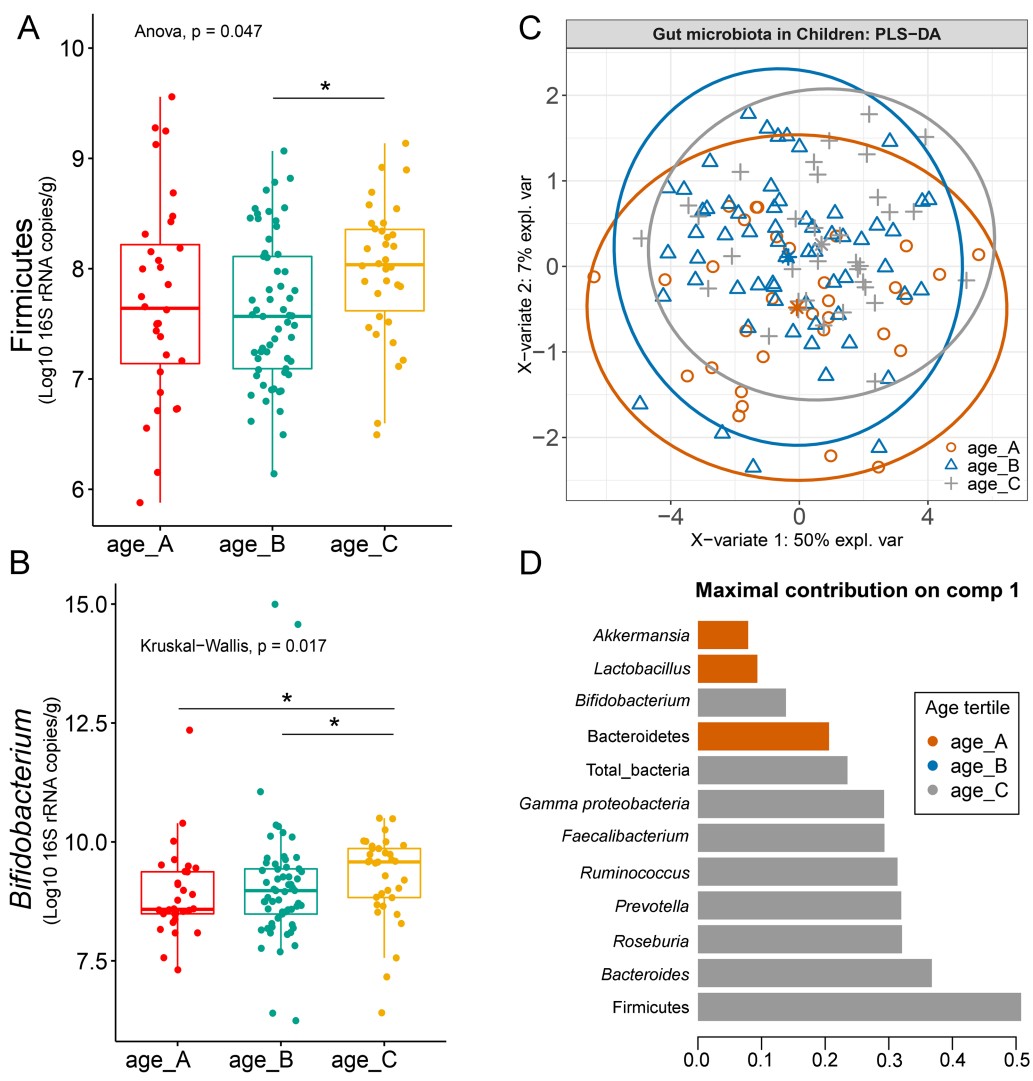

**Figure 5** **Partial least squares discriminant analysis (PLS-DA) of gut microbiota in school-aged children with different age tertile (corresponding to 25%, 50%, and 75%); age_A ≤ 8.05 years, age_B 8.05 < age < 11.06 years, age_C ≥ 11.06 years.** (A–B) Boxplots showing normalized bacterial abundances based on log10 qPCR 16S rRNA copy number per gram of feces. An asterisk (*) indicates a significant difference in microbiota abundance among feeding types (*q < 0.05, Tukey's HSD test and Dunn's test with Benjamini-Hochberg *p*-value correction method). (C) The sample plot represents variations in gut microbiota profiles of children with different age tertile (95% confidence ellipses). An explained variance was based on X-variate (normalized bacterial abundances). (D) Discriminant analysis demonstrating variable selection (microbiota taxa) for which the median (method = 'median') is maximum in component 1 of the sample plot. Horizontal bars indicate each bacterial taxon assigned to age tertile and their length corresponds to the loading weight. The importance of the bacteria contributing to the dimension runs from the bottom to the top of the figure.

abundances observed in component 1 of PLS-DA plots of gender (Fig. S18A). Classification models further demonstrated that *Lactobacillus*, *Gammaproteobacteria*, and *Bacteroides* were the top three microbiota taxa associated with girls (Fig. S18B). Based on assessing the discriminative ability of these microbiota taxa for each class (categorical variables), the test

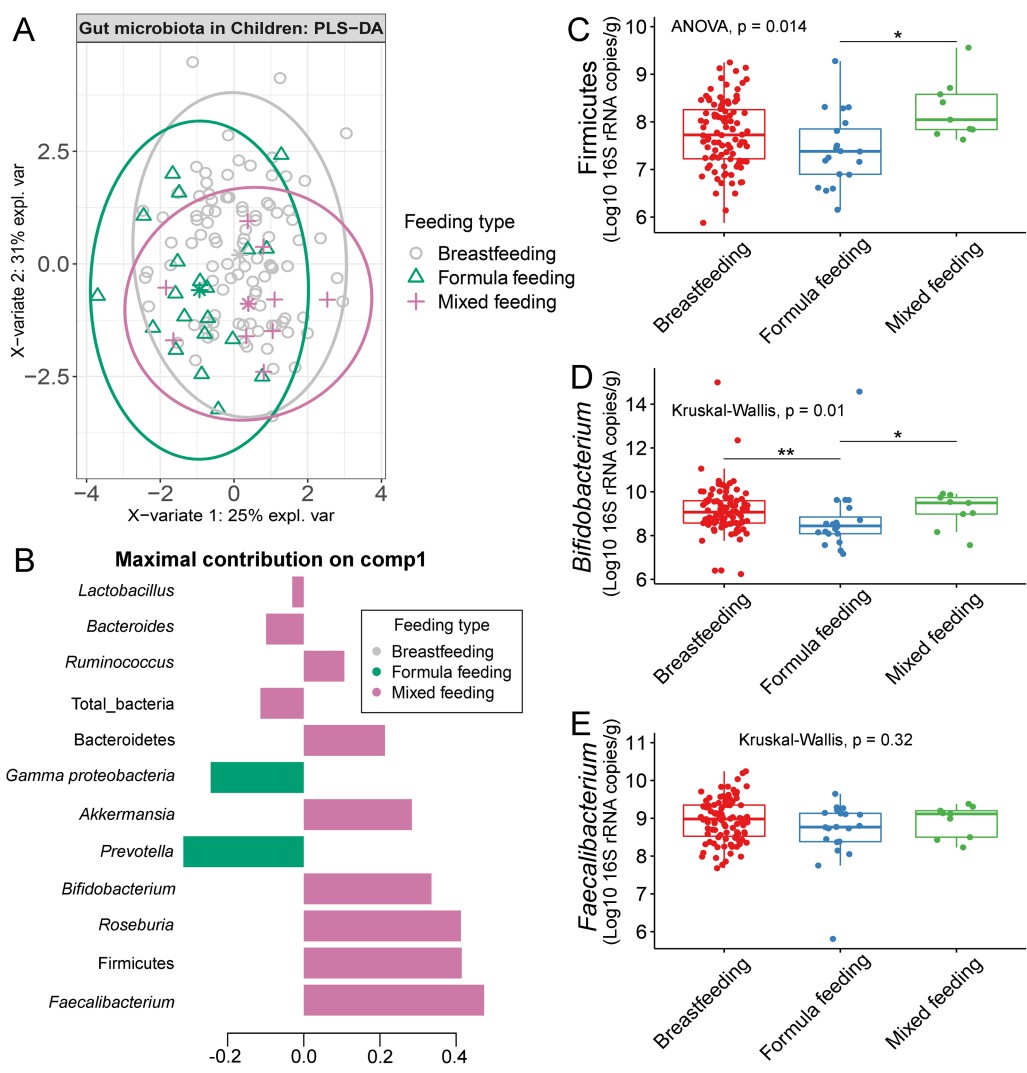

**Figure 6 Partial least squares discriminant analysis (PLS-DA) of gut microbiota in school-aged children with different feeding types (representing the feeding mode in infancy).** (A) The sample plot represents variations in gut microbiota profiles of school-aged children with different feeding types (95% confidence ellipses). An explained variance was based on X-variate (normalized bacterial abundances). (B) Discriminant analysis demonstrating variable selection (bacterial taxa) for which the median (method = 'median') is maximum in component 1 of the sample plot. Horizontal bars indicate each bacterial taxon assigned to feeding type and their length corresponds to the loading weight. The importance of the bacteria contributing to the dimension runs from the bottom to the top of the figure. (C–E) Boxplots showing normalized bacterial abundances based on log10 qPCR 16S rRNA copy number per gram of feces. Asterisks indicate a significant difference in bacterial abundance among feeding types (**$q < 0.01$, *$q < 0.05$, Tukey's HSD test and Dunn's test with Benjamini-Hochberg $p$-value correction method).

indicated that the outcome had poor discrimination capacity to distinguish between classes (AUC < 0.6, $p > 0.05$, Fig. S18C). The model indicated that gender did not influence the gut microbiota profiles of children in this study.

### Correlation between ethnicity and gut microbiota composition

No significant differences in the abundances of gut microbiota were found across ethnicity (Fig. S19). When we included ethnicity in the PLS-DA, the model demonstrated the association of this variable with the gut microbiota of children. While *Bacteroides* was the discriminative bacteria in Lahu ethnicity, *Gammaproteobacteria* was enriched in individuals of Akha ethnicity (component 1; AUC < 0.6, $p > 0.05$, Figs. S20A and S20B). However, a higher AUC value was obtained in component 2, where *Akkermansia* discriminated Thai Yai from others (AUC = 0.68, $p < 0.01$), while *Faecalibacterium* and *Roseburia* were the most discriminative bacteria in Akha ethnicity (AUC = 0.67, $p < 0.01$, Figs. S20C and S20D). These models implied that ethnicity had a slight influence on the gut microbiota of school-aged children.

## DISCUSSION

Our qPCR study of eleven microbiota taxa and total bacteria in the stool of 127 school-aged children revealed associations with dietary behaviors and demographic factors. Supervised analyses suggested that the gut microbiota profile was influenced by high fat foods consumption and the demographic factors of BMI $z$-score, age, mode of birth delivery, method of milk delivery, and ethnicity. Gender was not linked to variation in the gut microbiome in this study.

The human diet has a significant impact on the gut microbiota, as changes in the composition in response to food consumption have been extensively documented (*David et al., 2014*; *Zmora, Suez & Elinav, 2019*; *Leeming et al., 2019*). Here, we observed a significant increase in the abundance of *Gammaproteobacteria* in children who scored lower on high fat foods behaviors (characterized by the frequency of eating high fat foods, fried foods as well as food or dessert which was cooked with coconut milk, butter, or margarine). Previous studies in animals, and an *in vitro* model of the human gut using 16S rRNA gene sequencing, also showed that the abundance of bacteria belonging to the *Gammaproteobacteria* class increased following a high-fat diet consumption (*Lecomte et al., 2015*; *Agans et al., 2018*). To the best of our knowledge, a similar finding has not been previously reported based on qPCR. Whether decrease in abundance of *Gammaproteobacteria* is indeed associated with high fat diets in this population would require additional validation using larger sized cohorts, and ideally combination of both high throughput sequencing and qPCR for comparability across studies. An increased abundance of *Lactobacillus* and *Ruminococcus* were also associated with lower HFF consumption. The abundances of these bacteria are disturbed in animal models fed a high-fat diet (*Daniel et al., 2014*; *Resch et al., 2021*), which indicates that these taxa may not grow well in the gut in the presence of high-fat foods. Hence, in this study, lower reporting of HFF dietary behaviors among children perhaps favors the growth of these bacteria. Furthermore, reporting of high fruits and vegetables consumption appeared to be associated with total bacteria and *Prevotella*
levels. These results are consistent with previous studies of the influence of shifting from traditional to Western diets (high-fat/low-fiber) on the gut microbiota of Asian populations. *Prevotella*-type taxa were overrepresented in the gut of school-aged children in rural Thailand when compared with children in urban areas, who harbored more *Bacteroides*-type bacteria. Frequency of fruit and vegetable intake may therefore support different enterotypes, as was already reported for Filipino children from rural Baybay as well as for Thai vegetarians (*Ruengsomwong et al., 2014*; *Nakayama et al., 2015*; *Nakayama et al., 2017*). Outside of Asia, *Prevotella* dominates the microbiota communities of rural African children consuming diets high in fiber compared to those of European children (*De Filippo et al., 2010*). These converging findings emphasize the importance of a fiber-rich food diet to colonize the gut with *Prevotella* (*Kisuse et al., 2018*). High salty foods (HSF) intake affected the abundances of gut microbiota. Specifically, the butyrate producer *Faecalibacterium* and *Lactobacillus* were differentially associated with reported moderate to high salty foods and low salty foods consumption, respectively. A similar contrasting profile between *Roseburia* (another butyrate-producing bacterium) and *Lactobacillus* was previously shown in mice fed high- and low-salt diets. The former was enriched in mice fed high-salt diet (*Wang et al., 2017*), while the proportion of the latter was significantly reduced (*Wang et al., 2017*; *Miranda et al., 2018*). A similar finding has also been noted in humans (*Wilck et al., 2017*). These findings suggest that high salt food consumption impacts the abundance of specific gut microbiota members.

Changes in the gut microbiota profile of children have been associated with BMI status classified based on both centiles (*Bervoets et al., 2013*) and $z$-scores (*Golloso-Gubat et al., 2020*; *Shin & Cho, 2020*). In this study, a low abundance of Firmicutes and *Ruminococcus* was associated with obesity, while normal and overweight children had a high abundance of these bacteria. These findings are in contrast to previous studies based on 16S rRNA sequencing, whereby obese children had a high abundance of Firmicutes (*Da Silva, Monteil & Davis, 2020*), while *Ruminococcus* was nearly depleted in overweight/obese when compared to normal-weight children (*Karvonen et al., 2019*). A longitudinal study conducted in school-aged children with dietary records also highlighted a decrease of *Ruminococcaceae* in children who developed obesity and had a high calorie intake (high carbohydrate/high fat and high protein/high fat) associated with the obese status, (*Rampelli et al., 2018*). These findings suggest food intake and weight gain could contribute to variability in the gut microbiome (*Rampelli et al., 2018*). Despite unequal sample sizes and a different dietary assessment method herein, most obese children (72%) consumed high fat foods quite frequently (moderate to high risk) (Fig. S21), while only 33% ate fruits and vegetables (Fig. S22). Thus, the observed differences in microbiota abundance in our study were likely influenced by high-calorie diets, although further study with more participants, longer follow-up periods, and more extensive microbiome profiling is needed to verify this hypothesis.

The abundance of *Bifidobacterium* can vary across the stages of life (*Arboleya et al., 2016*; *Saturio et al., 2021*) and this genus is often enriched in the gut microbiota of children (*Derrien, Alvarez & De Vos, 2019*). A similar trend was also detected in our study with a high level of *Bifidobacterium* among school-aged children grouped by age tertile. The abundance

of *Bifidobacterium* was significantly high in children aged over 11 years. Moreover, children in previous studies that fell into the same age categories as in this study also had a higher fecal concentration of *Bifidobacterium* compared to those that were older aged (*Agans et al., 2011*; *Hollister et al., 2015*; *Zhong et al., 2019*). Concerning age variables, a gap may exist with these findings as we stratified individuals by quantile ranges. Whether or not the presence of this particular bacterium is associated with age, changes in *Bifidobacterium* levels from childhood to adolescence using narrow-age ranges may be worth investigating to better comprehend this relationship.

Both birth delivery method and feeding type appears to have a strong influence on the early-life gut microbiota (*Cukrowska et al., 2020*; *Mitchell et al., 2020*). The impact of the former has been shown in a large longitudinal analysis of gut microbiota from 600 newborns and 175 mothers, which denoted significant differences in the composition of gut microbiota between cesarean section born and vaginally delivered infants (*Shao et al., 2019*). The latter type of birth was associated with a high abundance of *Prevotella*, as shown in newborns and during the first two years of life (*Dominguez-Bello et al., 2010*; *Bokulich et al., 2016*). Although our study was conducted in school-aged children, enrichment of this genus was still observed in those who were born vaginally. This result implies that the impact of method of delivery may continue beyond infancy. Furthermore, we found that the abundance of *Bifidobacterium* was lowest in children who were formula fed as infants when compared with children who were either breast fed or mixed fed during infancy. *Bifidobacterium* abundance is increased in the gut of breast-fed infants rather than in those that are formula-fed. It has been speculated that the bacterium utilizes human milk oligosaccharides (HMO) (*Lee et al., 2015*; *Forbes et al., 2018*; *Lawson et al., 2020*). Our data suggests that a lack of exposure to breast milk at an early age may reduce abundance of gut *Bifidobacterium*, while mixed-feeding may stabilize the abundance close to breastfeeding. As time progresses, however, many other factors including the influence of one's diet is expected to also influence the makeup of one's gut microbiome.

Ethnicity introduces variations in the gut microbiota profiles through diet (*Khine et al., 2019*; *Dwiyanto et al., 2021*). Considering the small sample size of our study, however, our findings did not have an adequate power to identify the associations between the consumption of ethnic-based diets and the abundance of gut microbiota. We did, however, observe a trend when discriminating between ethnic groups. For instance, two genera within the phylum Firmicutes (*Faecalibacterium* and *Roseburia*) were associated with children of Akha ethnicity, whereas *Akkermansia* was mainly found to associate with the Thai-Yai ethnic group. These results are inconclusive due to a lack of dietary data relating to ethic cultural practices.

Although our study demonstrated the independent effect of each host factor on the gut microbiota, our results should be interpreted with caution. Major limitations include the lack of sample size estimation and data on cultural practices (*e.g.*, traditional diets, lifestyle, *etc.*). Since recruitment of subjects was based on voluntary participation, the number of subcategories was not homogeneous. In this regard, inter-individual variation was investigated using multivariate statistical analyses with all concerned factors. The same method has been implemented in our previous works (*Gruneck et al., 2020*;

*Chumponsuk et al., 2021*). Moreover, we were unable to collect data on cultural practices due to the language barriers, which might link to dietary behaviors of these school-aged children. Both limitations described above serve to limit our ability to explore correlations between important risk factors and the gut microbiome of school-aged children. One such potential confounding factor, physical activity, should also be included with future studies to better understand the role this plays together with BMI and diet.

## CONCLUSIONS

This study highlights how diet influences gut microbiota. A high abundance of *Gammaproteobacteria* was noted in children who reported the consumption of fewer high fat foods. Demographic factors such as BMI $z$-score, age, and feeding type also demonstrated their potential associations with gut microbiota. Obese children were characterized by a low abundance of *Ruminococcus*. Those over 11 years of age were found to have a high level of *Bifidobacterium*, whereas this abundance decreased in children with a history of formula feeding. Moreover, birth mode and ethnicity displayed a trend towards the enrichment of gut microbiota. Considering all host variables, gender was not a determinant of microbiota profiles in this study.

**Abbreviations**

| | |
|---|---|
| **HEB** | Healthy eating behavior |
| **FV** | fruits and vegetables |
| **HSFB** | high sugar foods and beverages |
| **HSF** | high salt foods |
| **HFF** | high fat foods |
| **OV** | overweight |
| **OB** | obese |
| **age_A** | age of children $\leq$ 8.05 years |
| **age_B** | age of children between 8.05 and 11.06 years (8.05 <years <11.06) |
| **age_C** | age of children $\geq$ 11.06 years |

## ACKNOWLEDGEMENTS

The authors would like to thank all volunteers for providing fecal samples and dietary data. We would like to thank Mr. Channarong Wanthanjai for technical assistance.

### Funding

This work was financially supported by Bangkok Dusit Medical Services through their corporate social responsibility support to Oregon Health & Science University (OHSU) and Mae Fah Luang University via the Gut Microbiome research group. The funders had no role in study design, data collection and analysis, decision to publish, or preparation of the manuscript.

## Grant Disclosures

The following grant information was disclosed by the authors:

Bangkok Dusit Medical Services through their corporate social responsibility support to Oregon Health & Science University (OHSU).

Mae Fah Luang University via the Gut Microbiome research group.

## Competing Interests

Kongkiat Kespechara is the founder of Sooksatharana (Social Enterprise) Co., Ltd.

## Author Contributions

- Lucsame Gruneck performed the experiments, analyzed the data, prepared figures and/or tables, authored or reviewed drafts of the paper, and approved the final draft.
- Eleni Gentekaki, Kongkiat Kespechara, Justin Denny, Lisa K. Marriott and Jackilen Shannon conceived and designed the experiments, authored or reviewed drafts of the paper, and approved the final draft.
- Thomas J. Sharpton analyzed the data, authored or reviewed drafts of the paper, and approved the final draft.
- Siam Popluechai conceived and designed the experiments, performed the experiments, analyzed the data, prepared figures and/or tables, authored or reviewed drafts of the paper, and approved the final draft.

## Human Ethics

The following information was supplied relating to ethical approvals (i.e., approving body and any reference numbers):

All participants provided written informed consent and the study was approved by the Ethics committee of Mae Fah Luang University (Ethics license: REH-61204). The study was conducted in accordance with the Declaration of Helsinki.

## Ethics

The following information was supplied relating to ethical approvals (*i.e.,* approving body and any reference numbers):

The study was approved by the Ethics committee of Mae Fah Luang University (Ethics license: REH-61204).

## Data Availability

The raw measurements are available in the Supplementary Files. The data were used for statistical analysis to determine the influence of dietary behaviors and demographic factors on the gut microbiota.

## Supplemental Information

Supplemental information for this article can be found online at http://dx.doi.org/10.7717/peerj.13325#supplemental-information.

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
