# Peer review of "The fecal microbiota of Thai school-aged children associated with demographic factors and diet"

_PeerJ, doi:10.7717/peerj.13325_

## Round 0.1 · original submission · Major Revisions

I have made some comments for style that may improve the text (see below). Importantly, since one of the two reviewers recommended rejection, I will need to find a second reviewer, if you choose to resubmit.

Line 39 & 40. Delete phrases like “have been shown to” and “are thought to” (see line 85 too).
Line 45. Delete “techniques”
Line 49. Delete “We found that”
Line 52. Present age of children in integers. In other words, there is no significant difference between a cohort of 11.06 year old children and 11 year olds. I suggest using traditional categories, like preteen.
Line 53. The phrase “history of vaginal delivery” needs revision. I suggest something like “that were delivered vaginally”
Line 55. A verb is missing here. Revise to “nonetheless appeared significantly linked”
Line 63. Here and throughout, use active voice if possible. Revise to “Bacteria belonging to…dominate the gut microbiome early in life. See also line 78.
Line 97. Delete “is one of the potential factors which”
Line 99. Replace “seem to explain” with “explains,” unless you doubt the publications cited.
Line 529. Provide page numbers for references (see also 531, 534, 542…). Also, I suggest you look at this paper: https://bmcbioinformatics.biomedcentral.com/articles/10.1186/s12859-019-2744-2

Reviewer 1 ·

Basic reporting

The paper started off nicely, with an abstract and introduction that clearly stated the background and aim of the study. All of the supplementary files are provided, but there are far too many files and some files are unclear.
The result part is overly explained, with all of the statistics specifications used are mentioned. It is highly encouraged for the authors to do a major revision on regards to the result part. A more thorough comment on which result part should be revised can be seen in the attached document of this review.

Experimental design

It is easy to find the research question and the research question itself is well-defined and relevant. The experimental design is explained well and replicable. However, it is not necessary to re-mention the statistical analysis in the result part. If it is needed, the statistical analysis method can be added as a footnote in the figure or table to simplify the result part.

Validity of the findings

The provided data are robust and statistically well-defined. But it is quite hard to review as it mentions all statistical methods used that is better to mention them only once in the materials and methods part of the paper. The discussion part is missing some explanation for the results.
Conclusions answered the aim of the study and were well-stated. However, it is unnecessary to re-mention the analysis used.

Additional comments

The importance of the study is well-received by the reviewer and the result will be impactful for future studies. However, there are major revisions needed for this paper including missing parts, structural revision, and overly-explained statistic methods. The author should focus on the results that are very crucial and answer the research question. It is not necessary to explain all of the results obtained. In the end, the reviewer would like to appreciate the authors' hard work and encourage the authors in revising the manuscript.

Annotated reviews are not available for download in order to protect the identity of reviewers who chose to remain anonymous.

Reviewer 2 ·

Basic reporting

Submitted manuscript describes an analysis of fecal microbiota in a large cohort of children, and their associations with biometric and life-style parameters. An impressive amount of data was collected and processed. Strengthes of the manuscript include large experimentally obtained dataset, a large number of supplementary materials and figures, and inclusion of data. The analytical output is generally well described. However, I have very significant concerns regarding the validity of the data and the choice of multivariate methods. Specific issues are mentioned below:

Experimental design

More information is needed on qPCR raw data analysis and experimental design - how were raw values adjusted to account for unequal amount of starting gDNA copies? Equal amount of total gDNA added into each well does not equal the same number of starting copies as DNA integrity will not be identical among different samples.

The chosen primers have a nested design – e.g., Ruminococcus is in phylum Firmicutes, Prevotella is in Bacteroidetes. While this can be used if one analyzes results for each individual primer set separately, the values cannot be combined in multivariate analyses because they are not independent. I.e., values for Fimicutes will include Ruminococcus abundance, and so forth. Thus, when running PSL-DA analyses, for example, please use variables at only one taxonomic level, e.g., genus.

Why many demographic variables are arbitrarily (e.g., age < 8.05 years, why not 8.00 or 8.11 or …) split into groups? Why not use the full range of values and do a regression-type analysis which had better statistical power?
Please explain why MFA was chosen as the main multivariate analysis tool, rather than other, in my opinion, more appropriate methods such as RDA or CDA? What are MFA advantages that make it the ebst choice for your particular dataset structure?

Validity of the findings

The qPCR data is pretty unreliable: for example, from Figure S12 one can see that total amount of Prevotella (genus within phylum Bacteroidetes) is higher than actual total Bacteroidetes (the same is true for genera of Firmicutes). Were the actual amplification rates incorporated into the CT calculations? Which of the values should the reader trust? Because both cannot be true simultaneously.

Lines 2221-222: “Bacteroides was highly correlated with dimension 1” – indicates that association is obtained post-hoc. Lines 246-247: “In component 2, Lactobacillus and Ruminococcus were the most discriminative bacteria in HFF-low risk” - indicates that this is obtained during discriminant analysis and is part of the discrimination model. So, which is the correct description of what was done?

Additional comments

MFA figures contain circles that likely represent confidence clouds. This is, however, not described in the figure legends.
There is no taxon called Bifidobacteria, genus name is Bifidobacterium
Some variables are not well described – e.g., the cohort includes school-age children but there is a variable describing formula-fed vs breast-fed children. I assume this represents the feeding mode in infancy, not at the time samples were collected. Please state this clearly.
Many bee swarp plots do not have any explanation for the values (i.e., units) shown on the Y axis (e.g. Figure 2CDE)
Please use the same test for the same type of data – e.g., panel 2D shows Anova results, but 2E – Wlech test.

---

## Round 0.2 · Minor Revisions

I think the manuscript presents a rich data set and the science is sound. I have a number of edits for style, which I made directly in the attached pdf. Importantly, I think some of the supplemental figures are misnumbered. For example, line 358 directs the reader to S17 to show "differences between gender." I think the appropriate figure is S19. Please double check all figure numbers.

Reviewer 1 ·

Basic reporting

The reviewer would like to thank the authors for taking our suggestion in revising the manuscript. Most of all the comments has been thoroughly accepted and changes has been made. However, there are still some points that are need to be revised. First of all, some of the subheadings are still not yet corrected (refer to the document attached in this review). Then, limitations part(preferably after the discussion part) can be added to explain the lack of sample size calculation and that cultural practices data cannot be collected due to language barrier.

Experimental design

As previously mentioned, the sample size calculation was missing from the original manuscript. The author replied that the recruitment was based on voluntary participation; hence, no calculation was made. The magnitude base of mean difference was investigated using multivariate statistical analysis. Although it may be clear that this method was done for some, it would be best to provide supporting reference that also applied this method as well.

Validity of the findings

No comment

Annotated reviews are not available for download in order to protect the identity of reviewers who chose to remain anonymous.

---

## Round 0.3 · Minor Revisions

I was unable to obtain a reviewer for this version of the manuscript, as the previous reviewer has not responded to several requests. To provide timely processing of the manuscript, I have reviewed the manuscript. See the attached pdf for my suggestions.

---

## Round 0.4 · accepted · Accept

The manuscript received missed reviews in the first round but you did a good job of addressing those comments and then making minor revisions for style and format.